# The Association between Diabetes and Human T-Cell Leukaemia Virus Type-1 (HTLV-1) with *Strongyloides stercoralis*: Results of a Community-Based, Cross-Sectional Survey in Central Australia

**DOI:** 10.3390/ijerph19042084

**Published:** 2022-02-13

**Authors:** Mohammad Radwanur Talukder, Hai Pham, Richard Woodman, Kim Wilson, Kerry Taylor, John Kaldor, Lloyd Einsiedel

**Affiliations:** 1Baker Heart and Diabetes Institute, Alice Springs Hospital, Alice Springs, NT 0870, Australia; radwan.talukder@baker.edu.au (M.R.T.); honghai.pham.vn@gmail.com (H.P.); 2Flinders Centre for Epidemiology and Biostatistics, Flinders University, Adelaide, SA 5001, Australia; richard.woodman@flinders.edu.au; 3National Serology Reference Laboratory, Melbourne, VIC 3065, Australia; kimmwilson23@gmail.com; 4Poche Centre for Indigenous Health and Wellbeing, Alice Springs, NT 0870, Australia; kerry.taylor@flinders.edu.au; 5Kirby Institute, University of New South Wales, Sydney, NSW 2052, Australia; jkaldor@kirby.unsw.edu.au; 6Alice Springs Hospital, Alice Springs, NT 0870, Australia

**Keywords:** *Strongyloides*, HTLV-1, diabetes, anaemia, Australia

## Abstract

In central Australia, an area that is endemic for the human T-cell leukaemia virus type-1 (HTLV-1), the prevalence of *Strongyloides stercoralis* and its association with other health conditions are unknown. A cross-sectional community-based survey was conducted in seven remote Aboriginal communities in central Australia, from 2014 to 2018. All residents aged ≥10 years were invited to complete a health survey and to provide blood for *Strongyloides* serology, HTLV-1 serology and HTLV-1 proviral load (PVL). Risk factors for *Strongyloides* seropositivity and associations with specific health conditions including diabetes and HTLV-1 were determined using logistic regression. Overall *Strongyloides* seroprevalence was 27% (156/576) (children, 22% (9/40); adults (≥15 years), 27% (147/536), varied widely between communities (5–42%) and was not associated with an increased risk of gastrointestinal, respiratory or dermatological symptoms. Increasing age, lower HTLV-1 PVL (<1000 copies per 10^5^ peripheral blood leucocytes) compared to the HTLV-1 uninfected group and community of residence were significant risk factors for *Strongyloides* seropositivity in an adjusted model. A modest reduction in the odds of diabetes among *Strongyloides* seropositive participants was found (aOR 0.58, 95% CI 0.35, 1.00; *p* = 0.049); however, this was lost when body mass index was included in the adjusted model (aOR 0.48, 95% CI 0.48, 1.47; *p* = 0.542). *Strongyloides* seropositivity had no relationship with anaemia. Exploring social and environmental practices in communities with low *Strongyloides* seroprevalence may provide useful lessons for similar settings.

## 1. Introduction

Infection with the soil-transmitted helminth *Strongyloides stercoralis* is highly prevalent in tropical and subtropical regions of the world, including Australia [1,2]. Globally, an estimated 370 million people are infected with *Strongyloides stercoralis* [3]. In Australia, *Strongyloides* is endemic across the tropical north of the country where prevalence rates range from 10 to 60% [2]. The nematode has a complicated lifecycle in which rhabditiform larvae excreted in human faeces develop into infective filariform larvae capable of entering the human host by penetrating intact skin. Larvae then enter the venous blood and migrate to the small intestine by way of the lungs [4]. In contrast to other nematodes that infect humans, lifelong infestation can be maintained through an auto-infective life cycle in which larvae penetrate the intestinal mucosa [4]. 

*Strongyloides* infection can cause diverse symptoms, which reflect the organs involved in the nematode’s life cycle. These include pruritus, gastrointestinal symptoms, such as abdominal pain and diarrhoea, and respiratory symptoms, such as cough and dyspnoea [5,6]. Immunocompromised individuals, particularly those treated with corticosteroids or infected with the human T-cell leukaemia virus type 1 (HTLV-1) [5,7,8,9], may be unable to control the larval burden [8,9], resulting in larval migration to the lungs and dissemination to organs, such as the central nervous system, which are not ordinarily affected by larval migration [8,9]. Larval penetration of the gut mucosa in such circumstances may result in life-threatening invasive infections with enteric bacteria [5]. Both corticosteroid therapy and HTLV-1 infection result in an impaired T helper type 2 (Th2) immune response, which is responsible for controlling many helminth infections including *Strongyloides* [10]. This is accompanied by downregulated mast cell degranulation, reduced eosinophil recruitment and impaired parasite-killing activity in those infected with HTLV-1 [7]. Consequently, the number of surviving larvae is increased in these individuals, and this increases the risk of hyperinfection and dissemination [7]. HTLV-1 infection has been associated with two-fold increased risk of symptomatic strongyloidiasis compared to patients who are HTLV-1 uninfected [11]. *Strongyloides* coinfection of people with HTLV-1 induces polyclonal expansion of HTLV-1-infected cells by activating the IL-2/IL-2R system, which could increase the risk of developing HTLV-1 associated diseases [12].

In a central Australian case series, seven of eleven patients with complicated strongyloidiasis were HTLV-1 seropositive, and four died [5]. HTLV-1 co-infection is also associated with the failure of treatment for strongyloidiasis and earlier progression to fatal adult T-cell leukemia [13,14]. Reduced efficacy of anthelmintic drugs against *Strongyloides* is a consequence of impaired Th-2 immune response, which is modulated by the high IFN-γ levels observed in HTLV-1 infected individuals [7]. Treatment failure with anthelmintic drugs may, therefore, be indicative of HTLV-1 infection [2]. A higher number of HTLV-1 infected cells in peripheral blood (HTLV-1 proviral load, PVL) is associated with a higher larval burden as defined by the presence of *Strongyloides* larvae in stool [15]. Collecting and processing stool samples is challenging in remote areas and strongyloides serology is increasingly used in this setting. Whether higher HTLV-1 PVL are associated with an increased risk of symptomatic strongyloidiasis when this is defined by strongyloides seropositivity has not been studied. 

The endemic areas of HTLV-1 and *Strongyloides stercoralis* overlap in Australia [2]. Relative to non-Indigenous Australians, Australian Aboriginal communities are disproportionately affected by both pathogens. The adult HTLV-1 prevalence in central Australia is the highest reported worldwide and approaches 40% [16], while the prevalence of strongyloides ranges from 21 to 60% in the tropical north of Australia [2,4,17]. Crowded and low-quality housing with inadequate sanitation underlie these high rates of infection with *Strongyloides* [1]. Notwithstanding the extremely high HTLV-1 prevalence in the arid interior of Australia [16] and the increased risks that strongyloides coinfection poses to people with HTLV-1, the epidemiology and clinical manifestations of *Strongyloides* have been rarely studied in central Australia. Community-based *Strongyloides* seroprevalence data have so far only been available from a single small community where 10 of 72 adults were *Strongyloides* seropositive [2,18]. Community-based data are needed to support the development of prevention and control strategies for strongyloidiasis in these communities [6]. 

A higher admission rate with strongyloidiasis was reported among adults with HTLV-1 in a retrospective cohort study in central Australia [19]. However, this finding may have been confounded by increased testing and treatment of patients known to be HTLV-1 infected [19]. Recent studies also suggest interactions between *Strongyloides* and other conditions. For example, strongyloidiasis was associated with an increased risk of anaemia in Ethiopia [20], and strongyloides seropositivity was associated with a reduced risk of diabetes among Aboriginal Australians who attended a remote health clinic [21]. 

We therefore collaborated with remote Aboriginal communities to determine whether *Strongyloides* seropositivity was associated with HTLV-1 infection and other health conditions, including diabetes and anaemia, in central Australia.

## 2. Materials and Methods

### 2.1. Study Setting and Population

Central Australia is an area of some 1,000,000 km^2^ in which most Aboriginal people live in isolated communities. English is generally the second or third language spoken in these communities. Residents of 7 remote central Australian Aboriginal communities (estimated resident population >15 years, 1153) [22] (designated alphabetically, A to G) participated in this cross-sectional study between August 2014 and June 2018. All communities were extremely disadvantaged as reflected by multiple socio-economic indicators, including median total family income per week (range AUD 492–840), average household size (range 3.7–5.2 members), low proportions of school attendance (range 19–40.3%), low levels of completed year 12 (range 4.3–9.4%), and high levels of unemployment (range 19–40.3%) [22,23].

Details of community engagement and the method of recruitment have been described previously [16]. Briefly, all Aboriginal community residents who were old enough to provide blood via venipuncture (generally ≥10 years) were eligible for recruitment. Participation was voluntary and informed written consent was obtained from all participants prior to commencement of health examinations and sample collections. Consent for children and adolescents (<18 years) was obtained from their legal guardians. Information about the project and for consent was provided in primary Aboriginal languages. A central survey site, usually the primary health care (PHC) clinic, was established in each community. Data from one community have been published previously [18].

### 2.2. Data and Specimen Collection

Participants were asked to complete a simple health questionnaire that recorded alcohol and tobacco consumption, and symptoms classified as respiratory (cough, wheeze and dyspnoea), dermatological (pruritus and rash) or gastrointestinal (abdominal pain and diarrhoea) [24]. The health survey was conducted by a specialist physician qualified in internal medicine and infectious diseases (LE) who was blinded to HTLV-1 and Strongyloides serostatus. An interpreter fluent in local primary languages assisted where necessary. Height and weight were recorded for a concurrent study of spirometry in settings where space could be provided by the participating PHC clinic [21].

Clinic records for five years prior to recruitment were subsequently reviewed for any evidence of comorbid conditions including smoking, harmful alcohol consumption, diabetes, hypertension, chronic kidney disease (CKD) and chronic liver disease. A diagnosis of diabetes was recorded if this was either stated in the medical record or a diabetic medication was prescribed.

Whole blood was collected from participants by venepuncture into EDTA tubes. Peripheral blood buffy coats (PBBC) and plasma were prepared and stored at −80 °C at Baker Heart and Diabetes Central Australia (Baker Institute) in Alice Springs.

### 2.3. Strongyloides Serology

Blood samples for *Strongyloides* serology were collected in gold top serum separator tubes (SST) and stored at 4 °C at the Baker Institute, Alice Springs. Samples were shipped to Western Diagnostic Pathology, Perth, for *Strongyloides* serology testing using a commercial IgG enzyme-linked immunosorbent assay (EIA-4208, DRG Instruments, GmbH, Germany), which detects IgG directed against the soluble fraction of filariform *Strongyloides stercoralis* larvae, in accordance with the manufacturer’s protocol.

### 2.4. Haematological Indices

Point of care tests (HemoCue^®^ WBC DIFF System and Hemocue Hb 301 system, Ängelholm, Sweden) were used to determine haemoglobin concentrations and the total and differential (neutrophils, lymphocytes, monocytes, eosinophils and basophils) white blood cells counts in peripheral blood. Peripheral blood eosinophilia was defined as >0.6 cells/10^9^/L.

### 2.5. HTLV-1 Tests

Buffy coats and plasma were shipped to the National Serology Reference Laboratory (NRL), Melbourne, in batches at intervals during the study. HTLV-1 status was confirmed by Western blot analysis (WB) (HTLV-I/II Blot 2.4, MP Diagnostics) on samples reactive on screening either by enzyme immunoassay (Murex HTLV-I + II, DiaSorin, Italy) or particle agglutination assay (Serodia HTLV-1, Fujirebio, Tokyo, Japan). HTLV-1 PVL was determined by polymerase chain reaction (PCR) using DNA extracted from PBBC cells. Primers and probes that targeted a highly conserved region at the 5′ end of the gag gene in the p19 coding region of HTLV-1c (Mel5; accession number, L02534) were expressed as HTLV-1 copies per 105 peripheral blood leucocytes (PBL). Consistent with previous studies [25], HTLV-1 PVL was stratified as low (<1000 copies per 10^5^ PBL) and high (≥1000 copies per 10^5^ PBL). The lower limit of detection was 6.5 copies for HTLV-1 (95% confidence interval [CI], 5.4–8.4) [18]. HTLV-1 infection was defined as a positive WB test or a positive HTLV-1 PCR test.

### 2.6. Ethics

The study was approved by the Central Australian Human Research Ethics Committee (HREC-14-242, 15-322, 16-384).

### 2.7. Statistical Analysis

For categorical variables, frequency and percentages were used; for continuous variables, mean, standard deviation (SD), and median were used. *Strongyloides* seropositive and seronegative groups were compared using Pearson’s χ^2^ test, Student’s *t*-test or Mann–Whitney test as appropriate for each variable. A multivariate logistic regression model was used to determine the factors associated with *Strongyloides* seropositivity (age, sex, residence, tobacco use, alcohol consumption, diabetes, chronic kidney disease stages 2–5, chronic liver disease, and HTLV-1 seropositivity). The odds ratio with 95% confidence intervals (CI) and *p*-value (<0.05) were used to report the associations between *Strongyloides* seropositivity and other variables. All analyses were conducted in Stata version14 (StataCorp, College Station, TX, USA).

## 3. Results

*Strongyloides* serology was tested for 576 participants, approximately 37.5% of the estimated resident population (children under 15, 40/169 (24%); adults (≥15 years), 536/1153 (46%)). Overall *Strongyloides* seropositivity rates for children and adults were 22% (9/40) and 27% (147/536), respectively.

Overall mean age of adults did not differ between *Strongyloides* seropositive (Strongyloides + ve) and seronegative (Strongyloides − ve) groups (Table 1). However, *Stron**gyloides* seroprevalence was highest among the 45–54 years group (40%) and lowest among the 35–44 years group (18%) (Table 1). *Strongyloides* seropositivity differed considerably between communities, ranging from 5% in community C to 42% in Community A and Community G (*p* < 0.001) (Table 1).

### 3.1. Clinical Findings

Prevalence of gastrointestinal, respiratory and dermatological symptoms did not differ according to *Strongyloides* serostatus (Table 2).

Though still in the normal range, neutrophil counts were significantly lower and eosinophil counts were significantly higher among *Strongyloides* seropositive adults (Table 3). There were no differences in haematological parameters among children according to *Strongyloides* serostatus (Table 3). No significant difference in mean haemoglobin level between *Strongyloides* seropositive and seronegative groups was observed for either adults or children (Table 3).

### 3.2. Strongyloides Seropositivity and Diabetes

The prevalence of comorbid conditions including diabetes and harmful alcohol consumption did not differ according to *Strongyloides* seropositivity (Table 2). Diabetes was associated with a modestly lower odds of *Strongyloides* seropositivity in the adjusted model (aOR 0.58, 95% CI 0.35, 1.00; *p* = 0.049) (Table 4). Significantly lower odds was also apparent for the *Strongyloides* seropositive group when risk factors for diabetes were analysed separately (aOR 0.59, 95% CI 0.37, 0.95; *p* = 0.03) (Table 4); however, there was no association between diabetes and *Strongyloides* seropositivity when BMI was included in the analysis (*n* = 379; aOR 0.48, 95% CI 0.48, 1.47; *p* = 0.542) (Table 4).

### 3.3. Strongyloides Seropositivity and HTLV-1

*Strongyloides* seroprevalence was higher in the HTLV-1 uninfected group (102/335, 30.4%) relative to the HTLV-1 infected group (45/201, 22.4%) (*p* = 0.043) (Table 2). When *Strongyloides* seroprevalence was stratified by HTLV-1 PVL, compared to the HTLV-1 uninfected group, HTLV-1 low PVL (20.8%) and HTLV-1 high PVL (25.4%) groups had lower *Strongyloides* seroprevalence, but it was not statistically significant (Table 2).

In an adjusted model, risk of *Strongyloides* seropositivity increased with age, and was reduced among participants with a low HTLV-1 PVL compared to HTLV-1 uninfected group (Table 5). Relative to residents of community C, risk of *Strongyloides* seropositivity was increased three to thirteen-fold among residents in other study communities (Table 5).

## 4. Discussion

In the first study of *Strongyloides* seroprevalence in remote central Australian communities, and the first to stratify risk according to HTLV-1 PVL, we found an overall prevalence exceeding 20% for Aboriginal adults and children. The nematode is most prevalent in the tropical and sub-tropical regions of developing countries [1], and high endemicity has long been recognised in remote Aboriginal communities in tropical Australia where prevalence rates may be as high as 60% [2]. Chronic exposure to *Strongyloides* reflects a persistent social disadvantage in remote Aboriginal communities in which overcrowded housing and non-functioning health hardware lead to environmental conditions that are detrimental to householders [1]. Although data on health infrastructure were not available, other socio-economic indices were substantially lower than national averages for all study communities [22,23,26]. *Stronglyoides* seroprevalence varied widely between communities in central Australia. Interestingly, the lowest seroprevalence was recorded for community C, which also had the largest households, lowest household income and lowest educational attainment according to ABS and regional data [23,26]. Identifying factors that protect these residents from the high regional prevalence of *Strongyloides* may contribute to the development of culturally appropriate control strategies elsewhere.

The mass drug administration (MDA) of ivermectin, either alone [27] or in combination with community participation [17] and environmental measures [6], has produced substantial reductions in *Strongyloides* prevalence in Australia [17,28] and overseas [29]. In one community in the far north of Australia, an ivermectin MDA reduced *Strongyloides* seroprevalence from 21% at baseline to 2% at month 18 [17]. Community ownership and improved sanitation are likely to be necessary for sustained reductions in prevalence [30]. Nevertheless, evidence of adverse outcomes among community residents who are *Strongyloides* seropositive is lacking and there has been considerable debate as to whether public health strategies to reduce *Strongyloides* prevalence are warranted in Australia [31], and no global consensus has emerged [32]. Although respiratory (16%), gastrointestinal (8%) and dermatological symptoms (8%) were relatively common in our cross-sectional study in Central Australian Aboriginal communities, these were not correlated with *Strongyloides* seropositivity. Similarly, a case–control study [31] and baseline and follow up clinical assessments for an MDA [17] in northern Australia were unable to find an association between *Strongyloides* seropositivity and clinical symptoms. Although there is currently no evidence that *Strongyloides* seropositivity is associated with gastrointestinal, respiratory or dermatological symptoms in remote Aboriginal communities [31] in the NT, correlating *Strongyloides* seropositivity with uncomplicated chronic strongyloidiasis in a population-based survey is difficult.

Notwithstanding reports that chronic helminth infections have a protective effect on metabolic diseases including reduced risks of hyperglycaemia, type 2 diabetes, metabolic syndrome and insulin resistance [33], the evidence for an association between *Strongyloides* and diabetes is scarce and remains inconclusive. A reduced risk of diabetes has been reported among Aboriginal Australians with *Strongyloides* infection in a community clinic setting [21] and in a hospital-based study from India [34]. Three principal mechanisms have been proposed to account for this protective effect [33]. A nutrition-based mechanism postulates that low body weight as a consequence of chronic helminth infection could contribute to improved metabolic outcomes [33]. Consistent with previous studies [35], we found no difference in BMI according to *Strongyloides* seropositivity and there was no association between *Strongyloides* infection and diabetes in the adjusted multivariate model that included BMI. Alterations to the gut microbiome of the human host are purported to result from helminth infection, and these may reduce the risk of diabetes by modulating glucose uptake, inflammation, and insulin sensitivity [33,36]. However, the relationship between helminths including *Strongyloides*, the microbiome and metabolic outcomes is yet to be fully explored [33].

A stronger case has been made for *Strongyloides*-mediated immune-modulation of glycaemic, hormonal, and cytokine parameters which may have anti-diabetogenic effects. Significantly reduced insulin and glucagon levels have been reported among *Strongyloides* infected individuals, which was confirmed by faecal microscopy, compared to those who are uninfected, and these effects reversed after anthelminthic therapy [36]. *Strongyloides* infection significantly reduces the pro-inflammatory milieu in T2DM by lowering the systemic levels of cytokines and chemokines [37]. Decreased blood insulin levels may result from an imbalance of pro- and anti-inflammatory adipokines. Adiponectin is a major inflammatory modulator that influences glucose homeostasis and insulin resistance in diabetes [38]. Modulation of adipocytokines due to *Strongyloides* infection may also confer a degree of protection against the severity of type 2 diabetes [36]. Chronic helminth infections mediate a modified Th2 immune response, resulting in reduced pro-inflammatory cytokines, such as IFN-γ and TNF-α, and an increase in anti-inflammatory cytokines, such as IL-10 and TGF-β [36]. An epiphenomenon of this survival strategy is likely to be an overall reduction in systemic inflammation within the host and an increase in insulin sensitivity [21]. This hypothesis is consistent with the low-level chronic inflammation that is a common feature of diabetes [39] and with the negative association between IL-10 and diabetes that has been reported in other settings [38]. Nevertheless, case–control studies from the UK [35] and Brazil [40] found an increased risk of *Strongyloides* seropositivity among adults with diabetes, and in the first community-based study to address this issue, we found that *Strongyloides* seropositivity was not associated with diabetes when BMI was included in the multivariate model.

Studying the health interactions of *Strongyliodes* in the human host is made challenging by the difficulty with which chronic strongyloidiasis is diagnosed and by the potential for *Strongyloides* serology to remain positive after resolution of infection. Although the identification of larvae in faecal samples by microscopy is considered the ‘gold standard’ method by which *Strongyloides* infection is diagnosed [41], the sensitivity of this technique is poor due to low and intermittent larval output in faeces [42]. In contrast, serological tests are highly sensitive, but have a lower specificity than faecal microscopy [43]. A recent systematic review and meta-analysis reported that the sensitivity and specificity of serological techniques relative to faecal microscopy also vary considerably for different tests, ranging from 59.1% to 90.5% and 61.8% to 93.7%, respectively [44]. The ELISA-IgG test that was used for *Strongyloides* diagnosis in this research has been reported as 92.3% sensitivity and 97.4% specificity relative to faecal results [42]. Cross-reactivity with other helminths, such as hookworm, can also result in false-positive results and overestimate the prevalence of *Strongyloides* [45]; however, this is unlikely to affect our results because nematodes other than *Strongyloides* are not endemic in central Australia. An underestimation of prevalence may also occur because of false-negative results, especially in acute infections and among immunosuppressed patients [43]. *Strongyloides* seroprevalence in HTLV-1 endemic areas, such as central Australia, may be underestimated by lower *Strongyloides* serology titres among individuals with HTLV-1 [2], and this could contribute to the lack of an association between HTLV-1 and *Strongyloides* seropositivity that is reported here and in other HTLV-1 endemic areas.

Several studies have reported an association between *Strongyloides* infection and anaemia [20,46]. In a recent Ethiopian study that included other soil transmitted helminths including hookworm, the presence of *Strongyloides* larvae in stool was associated with a five-fold increased risk of anaemia [20], although it was unclear whether other helminth infections were included in the adjusted model. No relationship between strongyloidiasis and anaemia was found in a recent systematic review of pregnant women [47]. Confounders such as the precarious socioeconomic conditions in which participants lived, which increased risk of exposure to *Strongyloides* and to malnutrition, might be the more important cause of anaemia and complicate the interpretation of these studies [47]. In the present study, and in a previous study of childhood strongyloidiasis in central Australia [48], all participants lived in similarly disadvantaged circumstances with ready access to health clinics and there was no association between anaemia and *Strongyloides* seropositivity.

Higher HTLV-1 PVL increases risk of symptomatic strongyloidiasis when this is defined by faecal microscopy [15], and this inability to control the larval burden may be responsible for higher rates of treatment failure among patients with HTLV-1 infection when treated with thiabendazole, albendazole, or ivermectin [13,14]. Our observation that *Strongyloides* seropositivity rates were lower among participants with low HTLV-1 PVL relative to those with high HTLV-1 PVL was therefore surprising. The effect of HTLV-1 PVL on *Strongyloides* seropositivity has not been studied previously, and further studies are necessary to understand this finding. However, this might reflect the greater likelihood of treatment for strongyloidiasis among people who are HTLV-1 seropositive and higher rates of treatment failure in those who with high HTLV-1 PVL.

Strengths of the present study are the recruitment of participants in a community setting with blinding of both researchers and participants to *Strongyloides* and HTLV-1 serostatus. However, some important limitations must be recognised. We recruited less than half of all eligible community members, raising the possibility of selection bias, and the recruitment of participants with better health engagement who may be more likely to receive treatment for strongyloidiasis. This might affect estimates of *Strongyloides* seroprevalence, and true prevalence may be higher than that reported here. However, selection bias is unlikely to alter the interpretation of our clinical survey, which was blinded to *Strongyloides* and HTLV-1 serostatus. Second, our clinical findings relate only to *Strongyloides* seropositivity, which reflects exposure to *Strongyloides* rather than active strongyloidiasis and has no relationship to larval burden. Third, we found no evidence of an association between *Strongyloides* seropositivity and gastrointestinal symptoms, which can fluctuate over time and may not be captured in our cross-sectional study. Our inability to obtain faecal samples prevented us from estimating the *Strongyloides* larval burden and we were also unable to exclude other pathogens, such as giardia and cryptosporidium, which are important causes of gastrointestinal symptoms in our region. Nevertheless, the number of participants with such symptoms was low for both *Strongyloides* seropositive and seronegative groups. Finally, we were unable to collect data for other important community and environmental factors, such as health engagement and treatment, or behaviours that might increase exposure to *Strongyloides*, which could explain the variation in prevalence.

## 5. Conclusions

In conclusion, we demonstrate a high *Strongyloides* seroprevalence in an arid, HTLV-1 endemic region of remote Australia. Consistent with other recent studies from the far north of the NT [17,31], *Strongyloides* seropositivity was not associated with clinical symptoms in our cross-sectional study in this HTLV-1 endemic area, nor was it associated with diabetes or anaemia. The single case of complicated strongyloidiasis that has been reported from this region since awareness was raised among clinicians a decade ago [5] followed iatrogenic immunosuppression without prior testing for strongyloidiasis [49]. This supports the current approach of testing and treating patients with clinical symptoms along with those with immune deficiencies due to HTLV-1 or iatrogenic immunosuppression [31]. Although the overall *Strongyloides* seroprevalence in central Australia was high, the wide variation in prevalence suggests that some Aboriginal communities may have developed protective strategies that reduce risk of infection with soil-transmitted helminths. An in-depth exploration of social and environmental practices in low-risk communities may provide useful lessons for other remote Aboriginal communities to reduce overall risk of strongyloidiasis.

## Figures and Tables

**Table 1 ijerph-19-02084-t001:** Demographic characteristics of adult Aboriginal participants by *Strongyloides* serostatus (*n* = 536).

Characteristics	Strongyloides − ve (*n* = 389)	Strongyloides +ve (*n* = 147)	*p*-Value
Demographic			
Age (Mean (sd))	38.0 (14.7)	40.4 (15.4)	0.108
Age groups (years) (n (%))			0.006
15–24 years	79 (73.8)	28 (26.2)	
25–34 years	95 (75.4)	31 (24.6)	
35–44 years	91 (82.0)	20 (18.0)	
45–54 years	68 (60.2)	45 (39.8)	
≥55 years	56 (70.9)	23 (29.1)	
Sex (n (%))			0.858
Male	178 (72.9)	66 (27.0)	
Female	211 (72.3)	81 (27.7)	
Residence (n (%))			<0.001
Community A	47 (58.0)	34 (42.0)	
Community B	48 (67.6)	23 (32.4)	
Community C	105 (94.6)	6 (5.4)	
Community D	93 (63.3)	54 (36.7)	
Community E	52 (85.3)	9 (14.7)	
Community F	25 (78.1)	7 (21.9)	
Community G	19 (57.6)	14 (42.4)	

**Table 2 ijerph-19-02084-t002:** Health-related conditions of adult Aboriginal participants by *Strongyloides* serostatus (*n* = 536).

Health Related Conditions ^a^ *n* (%)	Strongyloides − ve (*n* = 389)	Strongyloides +ve (*n* = 147)	*p*-Value
Diabetes			0.294
Yes	146 (75.3)	48 (24.7)	
No	243 (71.1)	99 (28.9)	
Hypertension			0.548
Yes	96 (70.6)	40 (29.4)	
No	293 (73.2)	107 (26.7)	
CKD ^b^			0.724
Yes	100 (71.4)	40 (28.6)	
No	289 (73.0)	107 (27.0)	
Liver disease			0.419
Yes	15 (65.2)	8 (34.8)	
No	374 (72.9)	139 (27.1)	
Smoking			0.068
Yes	209 (76.0)	66 (24.0)	
No	180 (69.0)	81 (31.0)	
Alcohol ^c^			0.779
Yes	164 (72.1)	60 (21.9)	
No	225 (73.2)	87 (26.8)	
Clinical history/examinations			
GI symptoms ^d^			0.252
Yes	16 (64.0)	9 (36.0)	
No	299 (74.4)	103 (25.6)	
Resp symptoms ^e^			0.924
Yes	49 (74.2)	17 (25.8)	
No	266 (73.7)	95 (26.3)	
Skin symptoms ^f^			0.381
Yes	15 (65.2)	8 (34.8)	
No	248 (73.6)	89 (26.4)	
BMI ^g^ (Mean (sd))	28.9 (6.9)	28.4 (5.5)	0.511
Blood results			
HTLV-1 ^h^			0.043
Yes	156 (77.6)	45 (22.4)	
No	233 (69.5)	102 (30.5)	
HTLV-1 PVL ^i^			
Uninfected	233 (69.5)	102 (30.5)	0.101
HTLV-1 low PVL	103 (79.2)	27 (20.8)	
HTLV-1 high PVL	53 (74.6)	18 (25.4)	

^a^ Health related conditions identified from medical records. These included diabetes, hypertension, chronic kidney disease (CKD stage 1 and above), and liver disease other than chronic hepatitis B virus, smoking and harmful alcohol consumption. ^b^ CKD—Chronic Kidney Disease, CKD Stage 1 and above. ^c^ Harmful alcohol consumption documented in medical records. ^d^ Gastrointestinal symptoms (*n* = 427) include abdominal pain and diarrhoea. Diarrhoea was defined as loose stools, >3 times per day for >2 days in the preceding week. ^e^ Respiratory symptoms (*n* = 427) include cough, wheeze and breathing problem. ^f^ Skin problems (*n* = 360) include pruritus and skin rash excluding scabies. ^g^ BMI—Body mass index as kg per m^2^, *n* = 379, from clinical examination. ^h^ HTLV-1 infection was determined by Western blot in 193 subjects and by PCR in 8 subjects with indeterminate WB. ^i^ Low and high HTLV-1 PVL were HTLV-1 PVL <1000 and ≥1000 copies per 105 peripheral blood leukocytes, respectively.

**Table 3 ijerph-19-02084-t003:** Haematological parameters according to *Strongyloides* serostatus for adults and children.

Blood Results (Median [Range]) ^a^	Adults *n* = 446			Children *n* = 31		
	−ve ^b^ *n* = 327	+ve ^c^ *n* = 119	*p*-Value ^d^	−ve ^b^ *n* = 26	+ve ^c^ *n* = 5	*p*-Value ^d^
Haemoglobin (g/L) (*n* = 441)	138 [44–192]	139 [89–217]	0.602	127 [68–165]	136 [113–165]	0.608
White Blood Cells (10^9^/L)	8.6 [3.6–18.8]	8.4 [3.2–18.6]	0.094	8.5 [4.4–14.0]	10 [4.1–11.7]	0.271
Neutrophils (10^9^/L)	4.8 [0.6–15.2]	4.5 [1.3–13.6]	0.047	4.0 [1.4–8.5]	5.3 [1.8–5.4]	0.501
Lymphocytes (10^9^/L)	2.7 [0.6–8.8]	2.5 [0.8–5.9]	0.377	3.2 [1.6–6.2]	4.0 [1.6–4.6]	0.484
Eosinophils (10^9^/L)	0.2 [0–2.7]	0.3 [0–2.0]	0.024	0.45 [0.1–1.1]	0.4 [0.1–1.4]	0.871

^a^ Blood results are presented with Median [range]. ^b^ −ve, *Strongyloides* seronegative. ^c^ +ve, *Strongyloides* seropositive. ^d^
*p* value, based on rank sum test.

**Table 4 ijerph-19-02084-t004:** The relationship between *Strongyloides* seropositivity and diabetes in Aboriginal adults.

	All Participants (*n* = 536)						Participants with BMI ^d^ (*n* = 379)					
Diabetes	uOR ^a^	95% CI	*p*-Value	aOR *^b^	95% CI	*p*-Value	uOR ^b^	95% CI	*p*-Value	aOR ^b^**	95% CI	*p*-Value

Age ^c^ (years)	1.09	1.07, 1.10	<0.001	1.09	1.07, 1.11	<0.001	1.08	1.06, 1.10	<0.001	1.08	1.06, 1.10	<0.001
Male gender	0.69	0.48 0.98	0.042	0.79	0.50, 1.24	0.306	0.56	0.36, 0.87	0.009	0.67	0.38, 1.17	0.125
Harmful alcohol consumption	1.48	1.04, 2.12	0.03	1.41	0.91, 2.20	0.123	1.45	0.95, 2.23	0.087	1.51	0.90, 2.55	0.121
Smoking	0.81	0.57, 1.15	0.24	0.94	0.59, 1.51	0.814	0.82	0.53, 1.26	0.367	1.30	0.74, 2.28	0.369
Strongyloides seropositive	0.81	0.54, 1.20	0.295	0.59	0.37, 0.95	0.03	0.92	0.57, 1.50	0.749	0.84	0.48, 1.47	0.542
BMI ^d^ (kg/m^2^)	-	-	-	-	-	-	1.09	1.05, 1.13	<0.001	1.07	1.03, 1.12	0.001

* Adjusted for age, sex, smoking, harmful alcohol consumption. ** Adjusted for age, sex, smoking, harmful alcohol consumption and BMI (body mass index). ^a^ unadjusted odds ratio. ^b^ Adjusted odds ratio. ^c^ Age per year. ^d^ BMI- Body Mass Index.

**Table 5 ijerph-19-02084-t005:** Risk factors for *Strongyloides* seropositivity among adult participants (*n* = 536).

Characteristics	Unadjusted			Adjusted ^a^		
	OR	95% CI	*p*-Value	OR	95% CI	*p*-Value
Age ^b^	1.01	0.997–1.02	0.122	1.02	1.005, 1.04	0.01
Male	0.96	0.66, 1.41	0.858	1.28	0.81, 2.01	0.291
Residence						
Community C	ref					
Community A	12.66	4.98, 32.20	<0.001	12.74	4.94, 32.85	<0.001
Community B	8.38	3.21, 21.93	<0.001	7.72	2.92, 20.39	<0.001
Community D	10.16	4.18, 24.70	<0.001	9.84	4.00, 24.18	<0.001
Community E	3.03	1.02, 8.96	0.045	2.93	0.97, 8.82	0.056
Community F	4.90	1.51, 15.86	0.008	3.89	1.16, 13.06	0.028
Community G	12.89	4.40, 37.74	<0.001	13.07	4.36, 39.18	<0.001
Smoking ^c^	0.70	0.48, 1.03	0.069	0.75	0.47, 1.19	0.217
Alcohol ^d^	0.95	0.64, 1.39	0.779	0.95	0.60, 1.52	0.839
Diabetes ^e^	0.81	0.54, 1.20	0.295	0.58	0.34, 1.00	0.049
CKD ^e^	1.08	0.70, 1.66	0.724	1.07	0.62, 1.84	0.816
CLD ^f^	1.43	0.59, 3.46	0.421	1.55	0.58, 4.27	0.362
HTLV-1 status ^g^						
Uninfected	ref					
HTLV1 low PVL ^h^	0.60	0.37, 0.97	0.038	0.55	0.32, 0.95	0.031
HTLV1 high PVL ^h^	0.77	0.43, 1.39	0.394	0.81	0.42, 1.56	0.531

^a^ model was adjusted for age, sex, residence (community), smoking, alcohol consumption, diabetes, CKD stage 1 ad above, liver disease and HTLV-1 status by PVL. ^b^ Age per year. ^c^ Smoking status documented in medical records. Reference is no smoking. ^d^ Harmful alcohol consumption documented in medical records. Reference is no harmful alcohol consumption. ^e^ Diabetes and chronic kidney disease (CKD) identified from medical records. ^f^ CLD—chronic liver disease other than chronic hepatitis B virus. ^g^ HTLV-1 infection was determined by Western blot in 193 subjects and by PCR in 8 subjects with indeterminate WB. ^h^ Low and high HTLV-1 PVL were HTLV-1 PVL <1000 and ≥1000 copies per 105 peripheral blood leukocytes, respectively.

## Data Availability

These data relate to an Australian Aboriginal population and are culturally sensitive. Although they cannot be shared publicly for ethical reasons, data will be made available to researchers who have received written approval from the Central Australian Human Research Ethics Committee (contact via cahrec@flinders.edu.au).

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
