# Peer review of "The Association between Diabetes and Human T-Cell Leukaemia Virus Type-1 (HTLV-1) with *Strongyloides stercoralis*: Results of a Community-Based, Cross-Sectional Survey in Central Australia"

_ijerph, 2022, doi:10.3390/ijerph19042084_

Round 1
Reviewer 1 Report
Thank you for allowing me to review this manuscript. This manuscript entitled "The Association between Diabetes and Human T-Cell Leukemia Virus Type-1 (HTLV-1) with Strongyloides Stercoralis: Results of a Community-Based, Cross-Sectional Survey in Central Australia". It is an interesting project, with a very current theme, although it has several limitations that make it suitable for publication in this magazine. These limitations are detailed below:
1. In the introduction it would be interesting to emphasize more the justification of the importance of the topic of study and its actuality.
2. In the material and methods section, different important aspects are not reflected:
• It should be reflected if the participation of the patients was voluntary or if they were offered a document with the information and informed consent.
• On the other hand, it is not specified whether the questionnaires are validated, nor the psychometric properties of said questionnaires.
• Another aspect that would be important to point out in the material and methods section is the design of the study.
3. The conclusions are clear and precise, but it would be interesting to include a future line due to the importance and timeliness of the topic. In addition, it would be interesting to emphasize what the results provide in clinical practice.
Good job and congratulations on the excellent manuscript.
Reviewer 2 Report
Read with great interest the findings presented by the authors. Appreciate the contribution to the overall seroprevalence and comorbidities of Aboriginal populations in Australia.
Consider adding more information in the opening sentence about the high prevalence of strongyloidiasis in Australia. Most graphs do not list it as endemic or high prevalence outside of the furthest northern part of the country. It looks like there has been considerable research added to the topic recently and would be good to provide more education to that fact as a premise for your study.
Consider adding a sentence or two more additional information/clarification to the Introduction with basics of HTLV-1 and the known geographical overlap and interactions with Strongyloides (e.g. acceleration of HTLV-1 disease). Especially why the association with HTLV-1 and treatment failure of strongyloidiasis.
Consider adding information in the Introduction regarding use of immunosuppresants and development of hyperinfection/disseminated strongyloidiasis. There was brief mention in the conclusion section to this fact and should be brought into the body of the paper somewhere prior to discussion there.
Page 3, line 111 - consider elaborating on what was used to define "harmful alcohol consumption" in the clinic records. Was this just the healthcare provider's subjective interpretation? Seems a bit subjective without a definition.
Also consider splitting the sentence from lines 109-112. The way it reads now, it looks like the reviewers were blinded to HTLV-1 and Strongyloides serostatus as well as DM, HTN, CKD and CLD. But I think they were blinded only to HTLV-1 and Strongyloides and specifically looking for the other comorbidities?
Page 4, line 153 mentions that a multivariate logistic regression was used to assess associated factors specifically in adults. Was there another test used for those people aged 15-17? If not consider removing the "adult" from that line.
Page 6, table 4 - suprascript the "c" after age, first row
Hopefully you will continue this work with follow-up publications on your hypothesis generating finding of lower Strongyloides seropositivity in those with lower HTLV-1 PVL. :)
